# Antimicrobial resistance associations with national primary care antibiotic stewardship policy: Primary care-based, multilevel analytic study

**Ashley Hammond**[1]*, **Bobby Stuijfzand**[2], **Matthew B. Avison**[3], **Alastair D. Hay**[1]

1 Centre for Academic Primary Care, Bristol Medical School, University of Bristol, Bristol, England, United Kingdom, 2 Jean Golding Institute, Royal Fort House, University of Bristol, Bristol, England, United Kingdom, 3 School of Cellular & Molecular Medicine, University of Bristol, Bristol, England, United Kingdom

* ashley.hammond@bristol.ac.uk

**Data Availability Statement:** All antibiotic dispensing data is publicly available via NHS Digital (https://digital.nhs.uk/data-and-information/publications/statistical/practice-level-prescribing-data). Antibiotic sensitivity data was collected from

## Abstract

### Background

Recent UK antibiotic stewardship policies have resulted in significant changes in primary care dispensing, but whether this has impacted antimicrobial resistance is unknown.

### Aim

To evaluate associations between changes in primary care dispensing and antimicrobial resistance in community-acquired urinary *Escherichia coli* infections.

### Methods

Multilevel logistic regression modelling investigating relationships between primary care practice level antibiotic dispensing for approximately 1.5 million patients in South West England and resistance in 152,704 community-acquired urinary *E. coli* between 2013 and 2016. Relationships presented for within and subsequent quarter drug-bug pairs, adjusted for patient age, deprivation, and rurality.

### Results

In line with national trends, overall antibiotic dispensing per 1000 registered patients fell 11%. Amoxicillin fell 14%, cefalexin 20%, ciprofloxacin 24%, co-amoxiclav 49% and trimethoprim 8%. Nitrofurantoin increased 7%. Antibiotic reductions were associated with reduced within quarter same-antibiotic resistance to: amoxicillin, ciprofloxacin and trimethoprim. Subsequent quarter reduced resistance was observed for trimethoprim and amoxicillin. Antibiotic dispensing reductions were associated with increased within and subsequent quarter resistance to cefalexin and co-amoxiclav. Increased nitrofurantoin dispensing was associated with reduced within and subsequent quarter trimethoprim resistance without affecting nitrofurantoin resistance.

Bristol Royal Infirmary and Southmead Hospital Microbiology Laboratories between 2013 and 2016 calendar years for all E. coli confirmed urinary tract infections. As this data contains patient postcode information, which could be potentially identifiable, it is not possible to publicly share this dataset in full. However, a de-identified dataset has been deposited on the University of Bristol Research Data Facility. Researchers can request access to the dataset via the University website: http://www.bristol.ac.uk/staff/researchers/data/accessing-research-data/, dataset entitled 'Antimicrobial susceptibility data anonymised (04-2020).

**Funding:** Antimicrobial Resistance Cross Council Initiative supported by the seven research councils (www.mrc.ac.uk/amr). Grant reference is NE/N01961X/1.

**Competing interests:** The authors have declared that no competing interests exist.

## Conclusions

This evaluation of a national primary care stewardship policy on antimicrobial resistance in the community suggests both hoped-for benefits and unexpected harms. Some increase in resistance to cefalexin and co-amoxiclav could result from residual confounding. Randomised controlled trials are urgently required to investigate causality.

## Introduction

Antibiotic resistance is considered one of the greatest threats to public health in the UK and worldwide. Primary care is responsible for over 75% of all antibiotics prescribed, [1] and therefore an important contributor to antibiotic resistance in the community. [2, 3] Oral antibiotics profoundly affect bacteria in the lower gastro-intestinal tract, which are thought to be the main source of auto-infection in the urinary tract (UTI). [4] Antibiotic resistant UTIs last longer and are more expensive to treat than susceptible infections, [5] and can lead to life-threatening urosepsis. [6] UTIs are the most common confirmed bacterial infection managed in primary care. [7] Urine samples submitted for susceptibility testing provide an abundant and accessible source of information on resistance prevalence.

Numerous strategies have been developed to encourage improved antibiotic stewardship internationally in primary, [8] and secondary care. [9] Many assume that bacteria with resistance genes are less 'fit' than susceptible strains, [10] and therefore that reducing antibiotic exposure should reduce resistance. [11] Since 2014/15, the NHS England quality premium has incentivised the reduced primary care prescribing of co-amoxiclav (amoxicillin-clavulanate), cephalosporins and quinolones for any infection. [12] The English Surveillance programme for antimicrobial utilisation and resistance (ESPAUR) suggests that between 2015 and 2017 this was effective, [1] and not associated with any unintended consequences. [13] However, to our knowledge, there has been no investigation of the antimicrobial resistance impact of these changes.

This ecological study aims to investigate the relationship between primary care antibiotic dispensing and resistance in community-acquired urinary *Escherichia coli*, exploring trends over a four-year study period. The study period will allow us to observe whether practice-level reductions in co-amoxiclav, cephalosporin and quinolone dispensing have resulted in reductions in their respective resistance profiles.

## Materials and methods

### Data collection

**GP practices and antibiotic dispensing.** We selected all GP practices exclusively sending urine samples to two laboratories in South West England between 2013 (due to a change of computer system, we could not collect data any further back than this) and 2016. The total number of antibiotic items dispensed (both prescribed and collected at a pharmacy) for each practice were extracted from NHS Digital (https://digital.nhs.uk/prescribing). We collated monthly data between January 2013 to December 2016 and generated quarterly totals for the 20 most commonly dispensed antibiotics (see S1 Table), including those used for the treatment of a UTI. We generated quarterly totals for our analysis based on findings from previous studies related to persistence of resistance once it develops. [2, 14] From the same website, we also collected the total number of registered patients per quarter per practice and linked these to

our practice-level dispensing data. We collected the proportion of children aged under 5 years registered at each practice from the Public Health England Fingertips website. [15].

**Antibiotic resistance.** Microbiological data were collected directly from the two laboratories: the Bristol Royal Infirmary (Lab A) and Southmead Hospital (Lab B). Both laboratories used the British Society for Antimicrobial Chemotherapy guidelines for antibiotic susceptibility testing at the time urine specimens were tested. Resistance data for all urinary *E. coli* sourced from the GP practices were collected for: amoxicillin, cefalexin, ciprofloxacin, co-amoxiclav, nitrofurantoin and trimethoprim. Also available from the laboratories for each sample were patients' age, sex and partial postcode. From these we used the Index of Multiple Deprivation (IMD) 2015 [16],and the Rural Urban Classification 2011, [17] both generated from patient postcode information, to assign patient-level deprivation and rurality scores. We excluded urine samples submitted from hospital wards or from outpatient clinics, and non-*E. coli* UTIs, since other uropathogens are likely to have different resistance patterns due to the presence of intrinsic resistance mechanisms. [18] We removed duplicate isolates, defined as any urinary *E. coli* from the same patient with the same susceptibility pattern within 60 days. Microbiological data was linked to antibiotic dispensing data via each patient's primary care practice code.

**Data analysis.** As well as investigating the relationship between *E. coli* resistance to each antibiotic tested and dispensing of that antibiotic (so called 'drug-bug' pairs), we tested how resistance to each of the antibiotics was related to total antibiotic dispensing (see S1 Table). We further tested for the relationship between trimethoprim resistance and nitrofurantoin dispensing, whilst taking into account trimethoprim dispensing, since we hypothesised that increased nitrofurantoin might lead to reduced trimethoprim resistance. [19]

Data from both laboratories were combined for all analyses, though since Labs A and B did not routinely test against amoxicillin and cefalexin respectively until late 2014, these data were drawn only from the laboratory undertaking the testing.

Two sets of analyses were run for all drug-bug pairs. First, we tested for a quicker, within-quarter, relationship between practice-level dispensing and resistance from urine samples from the same practices. As a delay could exist in the effect of dispensing, a second set of analyses tested practice-level dispensing rates with subsequent quarter practice-level resistance.

**Multilevel models.** Since urine samples are not independent observations (urine samples from the same practice in the same quarter might be expected to correlate more strongly than urine samples from different practices or different quarters) we fitted multilevel logistic regression models to allow for hierarchical dependencies using R, [20] and lme4, [21] packages. [22] It further allows for predictor variables to be included in the model at the appropriate level. For example, in our study dispensing is a predictor that varies by practice by quarter, whereas patient age is a predictor that varies by urine sample.

For the within-quarter analysis, we accounted for the hierarchical structure of the data by including random intercepts on the practice-quarter and practice level (i.e. the model intercepts could vary by practice-quarter and practice, which means that we assume that the average resistance can vary by practice and quarter). Our main predictor was the number of antibiotic items dispensed per 1000 patients for a given practice in a quarter. The model included the following covariates: patient age in years (patient level), IMD 2015 scores (patient level), rural/urban 2011 classification (patient level), percentage of children under five registered at practice (practice level), and number of patients registered at practice (practice-quarter level). All continuous variables were grand-mean centred. Rural/urban 2011 classification was coded as a dummy variable with the rural classification as the reference category.

For the subsequent quarter analysis, we used the same model specifications, apart from dispensing rates which were dated one or more quarters back. As it is unclear after what delay

potential associations of prescribing on antibiotic resistance might become apparent, we initially fitted these models with different delays, i.e. prescribing rates from the previous quarter, from two quarters ago, and from three quarters ago. We then compared statistical model performance for each of these and interpreted the results of those models which came out favourably in this comparison. It was important to limit the number of analyses as much as possible, and given that there were only marginal differences between the different time points, the quarter closest to the current quarter only was selected, reasoning that the shorter the delay, the less likely it would be that other factors are influencing the relationship between dispensing and resistance. Performance of the model was evaluated by using 10-fold cross-validation (see S2 Table).

### Patient and public involvement

No patients were involved in setting the research question or the outcome measures, nor were they involved in developing plans for design or implementation of the study. No patients were asked to advise on interpretation or writing up of results. Results will be disseminated to relevant patient communities through news media.

## Results

### Practice-level antibiotic dispensing

Dispensing data were available for 163 primary care served by Lab A and Lab B for all study years. The primary care practices included an average registered population of approximately 1.5 million patients per study year across a wide range of urban and rural areas in the South West of England, namely Bristol, Bath, North Somerset, Somerset, South Gloucestershire and Wiltshire.

Between 2013 and 2016, there were reductions in dispensing for most antibiotics at practice-level (Fig 1), and total dispensing of all antibiotics per 1000 registered patients reduced by 11% (see S3 Table). Individually, co-amoxiclav reduced the most, by 49%; amoxicillin 14%; cefalexin 20%; ciprofloxacin 24%; and trimethoprim 8%. Nitrofurantoin dispensing increased by 7%.

### Prevalence of antibiotic resistance

Microbiology data was supplied for 163 primary care practices between 2013 and 2016 (see S4 Table). Overall, 152,704 *E. coli* urine samples were cultured. Among all urinary *E. coli*, 65% of patients were 51 years old or over, and 87% were female. Most patients lived in urban areas (81%), with more patients living in the least than more deprived areas (Table 1). S5 shows the total number of *E. coli* urine isolates tested against each antibiotic included in the study and the percentage of resistant isolates per year. Resistance was highest against amoxicillin (52%) and trimethoprim (36%).

### Primary analysis

**Practice level relationship between antibiotic dispensing and resistance, within and subsequent quarters.** Table 2 reports the results of our primary analysis of the relationship between antibiotic dispensing and resistance. Full results including all covariates included in each model are reported in S6 and S7. The odds ratios for all analyses represent the change in odds of resistance for a change of one dispensed item per 1000 patients, adjusted for age, deprivation (IMD 2015), urban versus rural classification, number of patients registered at primary care practice and the proportion of children under 5 years registered at primary care practice.

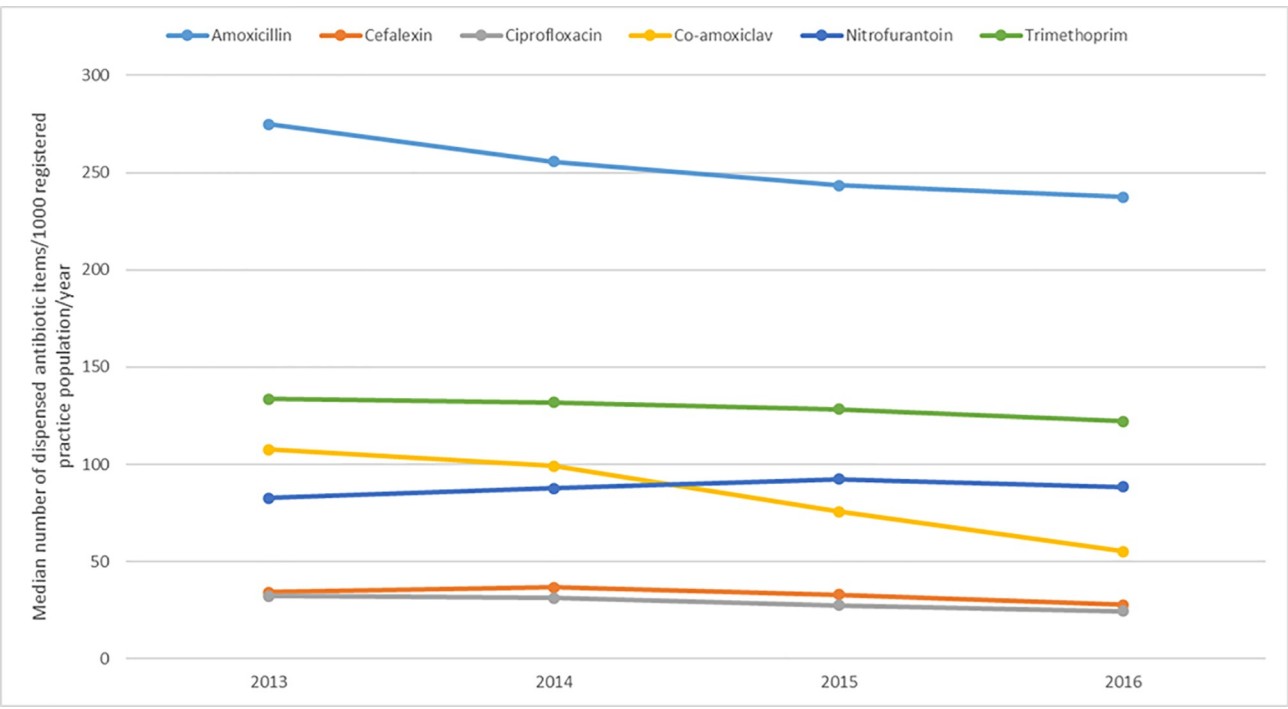

**Fig 1. Median number of dispensed antibiotic items per 1000 registered practice population per year.**

Table 1. Demographic characteristics of patients with E. coli UTI between 2013 and 2016.

| | Number (%) (N = 152,704) | Number (%) resistant ≥1 antibiotic (N = 78,987) | Number (%) sensitive to all antibiotics (N = 73,717) |
|---|---|---|---|
| **Age group:** | | | |
| <5 years | 3783 (2.48) | 1890 (2.39) | 1889 (2.56) |
| 5 to 15 years | 6718 (4.40) | 3079 (3.90) | 3643 (4.94) |
| 16 to 50 years | 43,463 (28.46) | 20,510 (25.97) | 22,948 (31.13) |
| >51 years | 98,733 (64.66) | 53,504 (67.74) | 45,234 (61.37) |
| Missing | 7 (0.00) | 4 (0.01) | 3 (0.01) |
| **Sex:** | | | |
| Female | 132,668 (86.88) | 68,316 (86.49) | 64,350 (87.29) |
| Male | 20,014 (13.11) | 10,655 (13.49) | 9361 (12.70) |
| Missing | 22 (0.01) | 16 (0.02) | 6 (0.01) |
| **Deprivation score:** | | | |
| (Least deprived) 1 | 45,431 (29.75) | 23,864 (30.21) | 21,570 (29.26) |
| 2 | 23,847 (15.62) | 11,244 (14.24) | 12,599 (17.09) |
| 3 | 38,472 (25.19) | 21,154 (26.78) | 17,167 (23.44) |
| 4 | 22,139 (14.50) | 11,509 (14.57) | 10,556 (14.32) |
| (Most deprived) 5 | 10,675 (6.99) | 5286 (6.69) | 5395 (7.32) |
| Missing | 12,140 (7.95) | 5930 (7.51) | 66210 (8.57) |
| **Urban/rural:** | | | |
| Urban | 123,087 (80.61) | 65,065 (82.37) | 58,021 (78.71) |
| Rural | 21,569 (14.12) | 10,374 (13.13) | 11,196 (15.19) |
| Missing | 8048 (5.27) | 3548 (4.49) | 4500 (6.10) |

**Table 2. Relationship between antibiotic dispensing and antimicrobial resistance within the same quarter[a].**

| | Reduced dispensing of same antibiotic[b] | | Reduced dispensing of all antibiotics[c] | | Increased dispensing of nitrofurantoin[e] | |
|---|---|---|---|---|---|---|
| | Odds ratio[d] | 95% CI | Odds ratio[d] | 95% CI | Odds ratio[e] | 95% CI |
| **Within quarter:** | | | | | | |
| Amoxicillin resistance | 0.998* | 0.850 to 0.972 | 0.999** | 0.998 to 1.000 | | |
| Cefalexin resistance | 1.033*** | 1.020 to 1.046 | 1.001 | 1.000 to 1.002 | | |
| Ciprofloxacin resistance | 0.982* | 0.965 to 0.999 | 0.999* | 0.998 to 1.000 | | |
| Co-amoxiclav resistance | 1.014*** | 1.008 to 1.019 | 1.000 | 0.999 to 1.001 | | |
| Nitrofurantoin resistance | 1.012 | 0.997 to 1.027 | 0.998 | 0.996 to 1.000 | | |
| Trimethoprim resistance | 0.996* | 0.992 to 1.000 | 0.999 | 0.999 to 1.000 | 0.991*** | 0.986 to 0.996 |
| **Subsequent quarter:** | | | | | | |
| Amoxicillin resistance | 0.997** | 0.995 to 0.999 | 0.999** | 0.998 to 1.000 | | |
| Cefalexin resistance | 1.033*** | 1.011 to 1.036 | 1.001 | 1.000 to 1.002 | | |
| Ciprofloxacin resistance | 0.982 | 0.965 to 1.000 | 1.000 | 0.998 to 1.001 | | |
| Co-amoxiclav resistance | 1.010*** | 1.004 to 1.016 | 1.000 | 0.999 to 1.001 | | |
| Nitrofurantoin resistance | 0.999 | 0.983 to 1.013 | 0.996*** | 0.994 to 0.998 | | |
| Trimethoprim resistance | 0.992*** | 0.988 to 0.997 | 0.999* | 0.999 to 1.000 | 0.994* | 0.989 to 0.999 |

Where

***p-value is <0.001

**p-value is <0.01

*p-value is <0.05

[a] Full model is presented in S6 and S7

[b] An odds ratio (OR) of <1 means reduced resistance associated with reduced dispensing, whereas an OR of >1 means increased resistance associated with reduced dispensing

[c] Total antibiotics includes: cefalexin, cefaclor, cefuroxime, azithromycin, clarithromycin, erythromycin, amoxicillin, co-amoxiclav, flucloxacillin, phenoxymethylpenicillin, ciprofloxacin, levlfloxacin, ofloxacin, doxycycline, lymecycline, tetracycline, trimethoprim, clindamycin, metronidazole, nitrofurantoin.

[d] Adjusted for age (years), deprivation (IMD 2015), urban versus rural classification, number of patients registered at primary care practice, and proportion of children under 5 years registered at primary care practice.

[e] Adjusted for trimethoprim dispensing, age (years), deprivation (IMD 2015), urban versus rural classification, number of patients registered at primary care practice, and proportion of children under 5 years registered at primary care practice.

These demonstrated associations at practice level between a lower rate of antibiotic dispensing and lower prevalence of antibiotic resistance (as indicated by odds ratios less than 1) for the following drug-bug combinations within quarter: amoxicillin, ciprofloxacin and trimethoprim. For reduced total antibiotics dispensed, we also found reduced within quarter resistance to amoxicillin and ciprofloxacin. In the opposite direction, we found lower rates of cefalexin and co-amoxiclav dispensing were respectively associated with increased cefalexin and co-amoxiclav resistance. A higher rate of nitrofurantoin dispensing was also associated with reduced prevalence of trimethoprim resistance. Concerning the covariates (see S6 Table and S7 Table), we consistently found that the odds of resistance increased with patient age and higher patient Index of Multiple Deprivation (IMD) scores. A higher percentage of children under five registered at the primary care practice increased odds of resistance for amoxicillin and co-amoxiclav. A greater number of patients registered at the primary care practice increased the odds of resistance to co-amoxiclav.

The next set of models that we fitted investigated the relationship between the rate of antibiotic dispensing in a given calendar quarter, and the prevalence of resistance in isolates cultured in the subsequent calendar quarter. Cross validation results are reported in S2 Table, which indicated that a delay of one quarter was either optimal or performed equally in the model with other delays for all dispensing-resistance combinations.

With the exception of the dispensing variables, model specifications remained the same as for within quarter models. These results indicate that relationships observed (in both directions) within quarter tended to persist to the subsequent quarter. Relationships between covariates and resistance we also remained similar to within quarter analyses (see S6 Table and S7 Table).

## Discussion

To our knowledge, this is the only study to evaluate the impact of recent English primary care stewardship policy on antibiotic resistance. In keeping with national trends, we found reductions in overall and individual antibiotic dispensing between 2013 and 2016, from 163 GP practices serving 1.5 million patients. [1] Antibiotic dispensing reductions were associated with reduced within quarter antibiotic resistance to ciprofloxacin, trimethoprim and amoxicillin, and these reductions persisted for three months for trimethoprim and amoxicillin. Of concern, some antibiotic dispensing reductions were associated with increased within and subsequent quarter resistance to cefalexin and co-amoxiclav. Reassuringly, nitrofurantoin (the go-to-first antibiotic for uncomplicated lower UTI) [23] dispensing increases were associated with reduced within and subsequent quarter trimethoprim resistance, without apparent changes in nitrofurantoin resistance.

The magnitudes of effect are both clinically and statistically significant. For example, the practice-level odds of resistance to trimethoprim decrease by 4% for every 100 fewer trimethoprim items dispensed per 1000 patients per annum, and the practice-level odds of resistance to cefalexin increase by 33% for every 100 fewer cephalosporin items dispensed per annum per 1000 patients.

### Strengths and limitations

Our study linked local practice-level antibiotic dispensing data with >150,000 routinely collected urine specimens testing positive for UTI caused by *E. coli*, which to our knowledge, is one of the largest of its kind, providing sufficient power to detect both within and subsequent quarter relationships between dispensing and resistance. Our study practices were found to be representative of both regional and national primary care practices. [15] Our study population however, included a slight over-representation of females, [24] and an under-representation of those living in the most deprived IMD quintile, [15] compared to regional and national averages. Our over-representation of females is nevertheless consistent with national primary care consultation data which suggests females consult more frequently than men. [25] The reasons for an under-representation of those living in the most deprived IMD 2015 quintile are largely unknown, but might relate to the inverse care law, where the availability of, and access to, medical care tends to vary inversely with the need of the population being served. [26] This is also supported by the fact that a recent survey on attitudes towards emergency care in England reported at 59% of those living in the most deprived IMD areas found it hard to get an appointment with their GP. [27] Therefore it may be that those living in the most deprived areas find it less easy to access their GP to provide a urine sample in the first instance.

The methods used in the study are robust; the multi-level modelling analysis enabled us to establish that associations between primary care antibiotic dispensing and resistance are independent of age sex, deprivation of study population, rurality, practice size and proportion of pre-school children registered at the practice. Further, as our antibiotic dispensing data was collected independently of our antibiotic sensitivity data from the laboratories, reporting of one was unlikely to have been influenced by knowledge of the other, adding to the reliability of our dataset. Also, although the use of fosfomycin is now encouraged for treatment of a UTI,

[23] it is rarely used in primary care, and so was not included in our top 20 most commonly prescribed primary care antibiotics. However routinely submitted urine specimens are not currently tested against fosfomycin for susceptibility, so it would not have been possible for us to determine prevalence of resistance. Finally, our study measures antibiotic use in number of items dispensed as opposed to prescribed. We consider this to be a much stronger measure of exposure and consumption since it reflects what patients have collected from a pharmacy and taken home. Unfortunately, it was not possible to collect routine data for antibiotic dispensing in secondary care, or ambulatory care, which are other potentially important sources of antibiotic consumption, due to its lack of availability as a routinely collected data source.

As with any population-based observational study, our findings do not provide information about individual patient risk of resistance, nor do the statistical associations observed mean the relationships are causal. Indeed, residual confounding from unmeasured variables, and/ or the ecological fallacy (e.g. individuals receiving fewer antibiotics might not be the ones experiencing UTIs with the use of some antibiotics being concentrated in higher risk patients, secondary care prescribing) could be operating. However, Bell and colleagues reported that antibiotic challenge at the population-level may be crucial in determining risk of resistance to antibiotics in the community, [28] and an advantage of this practice level analysis is that it inherently includes the hypothesised indirect effects of antibiotic exposure–i.e. if exposed individuals transmit resistant bacteria to unexposed individuals. Reverse causality seems improbable given the timing of dispensing reductions in relation to the timing of the NHS England quality premium.

## Results in the context of existing research

Our dispensing trends are similar to national trends reported in the 2018 ESPAUR report. This report indicated that nitrofurantoin consumption in England had increased by 28.8%, cefalexin consumption has decreased by 21.4%, and amoxicillin consumption had decreased by 7.4% between 2013 and 2017. [1] In 2014 nitrofurantoin was recommended as a first-line treatment for UTIs over trimethoprim, which likely accounts for the reductions in trimethoprim use we observed consistently between 2014 and 2016.

Previous studies, some now over 10 years old, have demonstrated compatible results to our study. Butler *et al* (2007) explored the relationship between ampicillin and trimethoprim dispensing and resistance in general practices in Wales, [5] as did Ironmonger *et al* (2018) on a wider array of drug-bug combinations. [29] However, unlike these studies, ours was conducted during a period of reducing overall antibiotic prescribing, and includes a broader range of antibiotics. Priest *et al* (2001) reported modest reductions in amoxicillin dispensing resulting in modest reductions in amoxicillin resistance. [30] This study however only explored this relationship over one year, and did not adjust for the use of other antibiotics, age, sex, rurality or deprivation. Pouwels (2018 and 2019) reported co-selection of resistance to antibiotics associated with prescribing, but did not adjust for possible confounding factors. [19, 31] No study has yet demonstrated the concerning rise in resistance we observed in relation to the decreased dispensing of some antibiotics. The explanation for this is not clear. The ecological fallacy could be operating as described above, or given people can become persistently colonised with resistant *E. coli* following a short stay in a different environment, for example during overseas travel or hospitalisation, [32, 33] another possibility is that co-amoxiclav and cefalexin resistance in the community is linked to use of these or related drugs in other settings, such as secondary care. In a 2017/18 survey of 900 cefalexin resistant urinary *E. coli* isolates from primary care in the same study area, 626 (69.6%) were found to be resistant to third generation cephalosporins used in secondary care, of which 571 (91.2%) produced the extended spectrum beta-

lactamase CTX-M. [34] Given that our study population is predominantly older in age, visiting, and even long-term use of healthcare facilities is possible.

We observed associations between a reduced rate of trimethoprim dispensing and a reduced likelihood of trimethoprim resistance, which was similar after one quarter than within the same quarter. This is comparable to the findings in a recent UK study investigating the association between use of different antibiotics and trimethoprim resistance. [19] This study noted that trimethoprim resistance could, in part, be explained by trimethoprim use in Enterobacteriaceae at the population level. As per our study, Pouwels *et al* also found reductions in trimethoprim resistance with increasing nitrofurantoin use. [19] This is expected given that nitrofurantoin has been recommended as the first-line treatment for UTIs over trimethoprim, therefore an increase in nitrofurantoin use, likely reflects a decrease in trimethoprim use, which are both used almost exclusively for the treatment of UTIs.

We were surprised that even at the practice level, associations were detectable within three months, and persisted for up to six months. These temporal relationships are comparable with those we have observed at the individual level. [2, 3] This is also consistent with an Israeli study where the prevalence of ciprofloxacin resistance was assessed before, during and after a nationwide restriction on quinolone use. [35] The study reported an immediate (same month) reduction in resistance levels. Another recent population-based study reported an association between higher rates of quinolone-resistant *E. coli* UTIs in populations with higher rates of quinolone prescribing, regardless of whether quinolones had been consumed by the individual patient. [36]

## Policy, clinical and research implications

Population, local and primary care practice level antibiotic stewardship policies based on 'first-principles' may result in both hoped-for benefits and unexpected harms. Our study suggests encouraging the first-line use of nitrofurantoin for uncomplicated lower UTI remains reasonable. Both policy makers and clinicians can be reassured that changes in dispensing can result in changes in resistance over a short timescale, but this also suggests national prescribing guidelines will need to be reviewed and updated frequently.

Given the concerning rise in cefalexin and co-amoxiclav resistance, practice level randomised controlled trials of prescribing guidance are needed urgently to establish causality. In our view, these should be part of a programme of real-time, one-health surveillance to improve our understanding of the vastly complex relationship between antibiotics, other factors, and resistance. Further community-based research is also needed to investigate these relationships at the individual patient level.

## Conclusions

This first evaluation of national primary care stewardship policy on community antimicrobial resistance suggests both hoped-for benefits and unexpected harms. The concerning increases in resistance to cefalexin and co-amoxiclav could be explained, at least in part, by residual confounding, and therefore require urgent investigation for causality in randomised controlled trials.

## Supporting information

**S1 Table. Antibiotic dispensing data collected from NHS Digital.**
(DOCX)

**S2 Table. Cross-validation results comparing models with different time delays for antibiotic resistance.** This procedure worked as follows: we trained the statistical model on 90% of the data (i.e. the training set), and then used this model to predict the remaining 10% of the data (i.e. the test set). The division between training and test set was made at random. A prediction accuracy, i.e. the percentage of cases in the test set for which the model predicted resistance correctly was then calculated. We repeated these two steps ten times, so that all observations had been part of the test set, and averaged the prediction accuracy over the ten test sets. This average prediction accuracy served as our criterion for statistical model performance. This procedure was repeated for all delays so that we could compare the statistical performance between the delays.
(DOCX)

**S3 Table. Median number of dispensed antibiotic items/1000 registered practice population/year.** Numbers in bold indicate the five largest relative decreases in antibiotic dispensing (%) between 2013 and 2016 [a] % decrease in dispensed items between 2013 and 2016 [b] negative numbers indicate an increase in antibiotic items dispensed between 2013 and 2016.
(DOCX)

**S4 Table. Total number of primary care practice urine samples received between 2013 and 2016.**
(DOCX)

**S5 Table. Number and percentage of resistant E. coli UTI per year.** [a] data for amoxicillin resistance from Lab B only [b] data for cefalexin resistance from Lab A only for 2013 and 2014, then Lab A and Lab B from 2015 to 2016.
(DOCX)

**S6 Table. Relationship between rate of antibiotic dispensing and prevalence of antibiotic resistance within the same quarter (full table of results).** Where [***]p-value is $<0.001$; [**]p-value is $<0.01$; [*]p-value is $<0.05$; IMD = Index of Multiple Deprivation 2015; Urban = Urban/ Rural Classification 2011 [a] The intercepts represent the average odds of observing resistance, keeping all else equal at the mean. i.e. an odds ratio of one indicates there is a 50% chance of observing resistance at the mean level of the covariates.
(DOCX)

**S7 Table. Relationship between rate of antibiotic dispensing and prevalence of antibiotic resistance in the subsequent quarter (full table of results).** Where [***]p-value is $<0.001$; [**]p-value is $<0.01$; [*]p-value is $<0.05$; IMD = Index of Multiple Deprivation 2015; Urban = Urban/ Rural Classification 2011 [a] The intercepts represent the average odds of observing resistance, keeping all else equal at the mean. i.e. an odds ratio of one indicates there is a 50% chance of observing resistance at the mean level of the covariates.
(DOCX)

## Author Contributions

**Conceptualization:** Ashley Hammond, Matthew B. Avison, Alastair D. Hay.

**Data curation:** Ashley Hammond, Matthew B. Avison.

**Formal analysis:** Ashley Hammond, Bobby Stuijfzand.

**Funding acquisition:** Matthew B. Avison, Alastair D. Hay.

**Investigation:** Ashley Hammond.

**Methodology:** Ashley Hammond, Bobby Stuijfzand, Alastair D. Hay.

**Project administration:** Ashley Hammond.

**Supervision:** Matthew B. Avison, Alastair D. Hay.

**Validation:** Ashley Hammond, Bobby Stuijfzand.

**Writing – original draft:** Ashley Hammond, Alastair D. Hay.

**Writing – review & editing:** Ashley Hammond, Bobby Stuijfzand, Matthew B. Avison, Alastair D. Hay.

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
