## [Decision Letter · Decision Letter 0]

8 Jan 2020

PONE-D-19-32146

Antimicrobial resistance associations with national primary care antibiotic stewardship policy: community -based, multilevel analytic study

PLOS ONE

Dear Dr. Hammond,

Thank you for submitting your manuscript to PLOS ONE. After careful consideration, we feel that it has merit but does not fully meet PLOS ONE’s publication criteria as it currently stands. Therefore, we invite you to submit a revised version of the manuscript that addresses the points raised during the review process.

Two reviewers have commented on the manuscript. There are major concerns about number of aspects, methodology, database used, selection of data and discussion. The authors need to address all comments point by point.

We would appreciate receiving your revised manuscript by Feb 22 2020 11:59PM. To enhance the reproducibility of your results, we recommend that if applicable you deposit your laboratory protocols in protocols.io, where a protocol can be assigned its own identifier (DOI) such that it can be cited independently in the future. For instructions see: http://journals.plos.org/plosone/s/submission-guidelines#loc-laboratory-protocols

We look forward to receiving your revised manuscript.

Kind regards,

Iddya Karunasagar

Academic Editor

PLOS ONE

Journal Requirements:

3. Please include a copy of Table 4 which you refer to in your text on line 184.

Additional Editor Comments (if provided):

Two reviewers have commented on the manuscript. There are major concerns about number of aspects, methodology, database used, selection of data and discussion. The authors need to address all comments point by point.

Reviewers' comments:

Reviewer's Responses to Questions

**Comments to the Author**

1. Is the manuscript technically sound, and do the data support the conclusions?

Reviewer #1: Partly

Reviewer #2: No

2. Has the statistical analysis been performed appropriately and rigorously? 

Reviewer #1: I Don't Know

Reviewer #2: I Don't Know

3. Have the authors made all data underlying the findings in their manuscript fully available?

Reviewer #1: Yes

Reviewer #2: No

4. Is the manuscript presented in an intelligible fashion and written in standard English?

Reviewer #1: Yes

Reviewer #2: No

5. Review Comments to the Author

Reviewer #1: Overall this is an interesting and exciting article, focusing on the all-important question of whether ambulatory antibiotic stewardship can impact community-wide antibiotic resistance

Minor comments: there seem to be some minor grammatical errors in the abstract's methods

The term drug-bug seems too informal for this work

I don't think you can say that antibiotic stewardship may lead to unanticipated harms based on the data you have presented. I think you might need to look at important confounders (movement in and out of the area, hospital prescribing practices, etc)

Methods: Did susceptibility testing cutoffs change during the period of the study?

Why were specimens excluded from those submitted from outpatient clinics? Wouldn't this be all of the specimens you were interested in studying?

The argument you give for excluding non-Ecoli specimens makes little sense to me. Yes, Enterobacter may produce amp-C, but then you'd say that you only want to exclude Enterobacter.

Now that I've read the methods, I see that I misinterpreted what you were trying to do. You were not trying to measure community-level resistance, but instead practice-level resistance. Make this clearer in the title, abstract, and hypothesis. I'm curious why the decision was made to try to analyze the data this way. Can patients go to other practices? Can plasmids be transferred from patient in one practice to a patient in another practice? Please explain why this decision was made.

Results:

I seem to be missing tables 3 and 4

At this point I seem to be unable to fully review the manuscript as it stands.

Discussion: The results really make no sense. Is there a reason why prescribing less cephalosporins in a practice is associated with a 33% increased odds of cephalosporin resistance? Again I couldn't review the tables so maybe I am missing something.

I am unclear what you mean in lines 228-229. Why would someone in one practice be more likely to transmit a drug resistant organism to another person in the same practice than, say, another person in the same town, or who works at the same location, or goes to the same school? Especially as repeat cultures from the same person with the same strain of organism are excluded. This makes no sense to me and is honestly one of the reasons I am struggling so much with this paper.

Other huge limitations to this sort of study are the lack of tracking non-E coli organisms, the lack of inclusion of antibiotic prescribing trends in hospitals, emergency departments, and other places where patients might receive antibiotics that would presumably also impact resistance, people moving in and out, etc. You do eventually get to this but highlight this early on as one of the major issues. And when you discuss the vastly increased rates of ESBL-producing Ecoli, this makes much more sense to me, as prevention of transmission of this strain may be impacted by isolation precautions, etc in the LTCF that you describe. I'm not sure I'd jump right to changing policies by avoiding amox-clav or cephalosporins, but instead determining which of these are related to CTX-M.

Reviewer #2: The authors have linked an impressive database of over 150,000 urine sample from UTI with the NHS database containing 163 GPs with 1.5 Mio registered patients.

This manuscript need major revisions. However, the authors have a riche database and I would like to encourage the authors to carefully address my inputs below.

Major concerns regarding the research aim:

- Line 19: you did not assess the association between national stewardship policy and antimicrobial resistance. You assessed the association of antibiotic reduction with antibiotic resistance. Please revise throughout your manuscript.

- Line 64: The design of the study is unclear: is this a before after analysis following an intervention (start of a NHS England quality premium) or is this a trend analysis of antibiotic prescribing and resistance over time. The introduction suggest the former the analytic approach more of the latter.

- The title of this manuscript suggests to investigate the impact of the national stewardship programme on AB prescribing patterns in selected practices in the southwest UK. What was the penetration of the programme in the area. How can we be assured that the programme did change prescribing behaviour. What was the type of the intervention? More details and references are needed here.

- In the method section a paragraph should be introduced that this study is limited to 163 pratices in the catchment area of the University of Bristol, UK and two labs serving the area. Were all practices selected in the reference area. Did the lab cover the whole area?

- Line 68 The time frame of the study should be better justified.

- You should restrict your analysis to antibiotics typically prescribed for UTI, as the lab-databases only contain resistance information from urine samples.

- The main problem of this study relates to the fact that only aggregated data on the practice level was analysed. Thus, changes in resistance patterns in urinary tract infections cannot be directly linked to changes in prescribing policy at the practice level as antibiotics may have been prescribed for other indications than urinary tract infections. This is a major drawback which should be better detailed in the discussion section.

- The manuscript lacks focus. The development of antibiotic use over time is shown for compounds that are not used for the treatment of urinary tract infections. Why should we be interested in marcrolid use for the management of urinary tract infections.

- We do not know anything about the case mix of patients in individual practices. This could be approximated for example by looking at co-medications that would allow to define some at risk populations like diabetic patients for URTI.

- We do not know anything about how complicated URTI or pyelonephritis were managed. Such data could have been derived from individuals practices of approximated by looking at hospitalization rates for these conditions during the respective observation periods.

- Why is fosfomycin not on the list of antibiotics?

- There seems to be a selection bias in the entire patient population at work that limits the generalizability of findings. URTI cultures appear to be more frequently done in less deprived patient populations in the UK. Any explanation for this?

Major concerns regarding databases:

The databases (especially from NHS) and the linkage process of the databases are poorly described and therefore, it is difficult to evaluate the analysis and the findings. However, to my understanding, the NHS database contains 1.5 Mio entries, whereas the lab-databases only 150,000 limited to UTI. Moreover, the NHS are probably aggregated per month (right?).

Therefore:

- Describe the databases, individual patient data or aggregated data?, available variable relevant for your analysis, for both (!) data sources

- How did you identify UTI patients in NHS database? If not (because data is aggregated), you need to say so.

- Describe linkage process, on which level where data linked, practice or patient-level? Linkage variables?

- Please state any data privacy concerns (concerns individual patients and practices), did you seek ethical approval? Was this study registered?

Major concerns regarding analysis:

- Line 70: You use monthly summary (aggregated data?!) to prepare quarterly totals? Then you analysis them per year? This needs clarification because with every summary step you lose information.

- Line 102 following: please specify your model: Logistic? Linear? Provide a reference for the chosen model and specify the statistical software used for the analyses.

- Random variation for quarterly analysis? I would expect seasonal variations, but those are not random. The rationale for the use of quarterly analysis is insufficiently specified. Other models modeling AB prescription and resistance over the entire observation period with spline functions might eventually be more efficient.

- Line 114 What are IMD 2015 scores exactly what do they best reflect?

- Line 128 Cross validation is insufficiently specified. How was this done by removing individual data points? Was complete pooling, no pooling and multilevel estimates compared. Which method was used (Price 1996)?

6. PLOS authors have the option to publish the peer review history of their article (what does this mean?). If published, this will include your full peer review and any attached files.

Reviewer #1: No

Reviewer #2: No

---

## [Author Response · Author response to Decision Letter 0]

5 Mar 2020

We have provided in our attached 'Response to Reviewers' document, a table detailing our response to each reviewer comment in detail, along with any changes made to the manuscript.

---

## [Decision Letter · Decision Letter 1]

24 Apr 2020

Antimicrobial resistance associations with national primary care antibiotic stewardship policy: primary care-based, multilevel analytic study

PONE-D-19-32146R1

Dear Dr. Hammond,

We are pleased to inform you that your manuscript has been judged scientifically suitable for publication and will be formally accepted for publication once it complies with all outstanding technical requirements.

With kind regards,

Iddya Karunasagar

Academic Editor

PLOS ONE

Additional Editor Comments (optional):

All reviewer comments addressed satisfactorily

Reviewers' comments:

Reviewer's Responses to Questions

**Comments to the Author**

1. If the authors have adequately addressed your comments raised in a previous round of review and you feel that this manuscript is now acceptable for publication, you may indicate that here to bypass the “Comments to the Author” section, enter your conflict of interest statement in the “Confidential to Editor” section, and submit your "Accept" recommendation.

Reviewer #3: All comments have been addressed

2. Is the manuscript technically sound, and do the data support the conclusions?

Reviewer #3: Yes

3. Has the statistical analysis been performed appropriately and rigorously? 

Reviewer #3: Yes

4. Have the authors made all data underlying the findings in their manuscript fully available?

Reviewer #3: Yes

5. Is the manuscript presented in an intelligible fashion and written in standard English?

Reviewer #3: Yes

6. Review Comments to the Author

Reviewer #3: (No Response)

7. PLOS authors have the option to publish the peer review history of their article (what does this mean?). If published, this will include your full peer review and any attached files.

Reviewer #3: Yes: Hare Krishna Tiwari

---

## [Editor Report · Acceptance letter]

4 May 2020

PONE-D-19-32146R1 

Antimicrobial resistance associations with national primary care antibiotic stewardship policy: primary care-based, multilevel analytic study 

Dear Dr. Hammond:

I am pleased to inform you that your manuscript has been deemed suitable for publication in PLOS ONE. Congratulations! Your manuscript is now with our production department. 

With kind regards,

on behalf of

Dr. Iddya Karunasagar 

Academic Editor

PLOS ONE